# Anxiety and Panic Buying Behaviour during COVID-19 Pandemic—A Qualitative Analysis of Toilet Paper Hoarding Contents on Twitter

**DOI:** 10.3390/ijerph18031127

**Published:** 2021-01-27

**Authors:** Janni Leung, Jack Yiu Chak Chung, Calvert Tisdale, Vivian Chiu, Carmen C. W. Lim, Gary Chan

**Affiliations:** 1National Centre for Youth Substance Use Research, The University of Queensland, Brisbane, QLD 4072, Australia; j.leung1@uq.edu.au (J.L.); c.tisdale@uq.edu.au (C.T.); vivian.chiu@uq.net.au (V.C.); c.lim2@uq.net.au (C.C.W.L.); c.chan4@uq.edu.au (G.C.); 2School of Psychology, The University of Queensland, Brisbane, QLD 4072, Australia

**Keywords:** COVID-19, pandemic, anxiety, panic buying, social media, psychological phenomena, snowball effect

## Abstract

*Background*: The coronavirus disease 2019 (COVID-19) pandemic had increased population-level anxiety and had elicited panic buying behaviour across the world. The over-hoarding of toilet paper has received a lot of negative public attention. In this work, we used Twitter data to qualitatively analyse tweets related to panic buying of toilet paper during the crisis. *Methods*: A total of 255,171 tweets were collected. Of these 4081 met our inclusion criteria and 100 tweets were randomly selected to develop a coding scheme in the initial phase. Random samples of tweets in folds of 100 were then qualitatively analysed in the focused coding phase until saturation was met at 500 tweets analysed. *Results*: Five key themes emerged: (1) humour or sarcasm, (2) marketing or profiteering, (3) opinion and emotions, (4) personal experience, and (5) support or information. About half of the tweets carried negative sentiments, expressing anger or frustration towards the deficiency of toilet paper and the frantic situation of toilet paper hoarding, which were among the most influential tweets. *Discussion*: Panic buying of toilet paper was seen during the 2020 pandemic period with a mass amount of related content spread across social media. The spontaneous contagion of fear and panic through social media could fuel psychological reactions in midst of crises. The high level of negative social media posts regarding the toilet paper crisis acts as an emotional trigger of public anxiety and panic. *Conclusions*: Social media data can provide rapid infodemiology of public mental health. In a pandemic or crisis situation, real-time data could be monitored and content-analysed for authorities to promptly address public concerns.

## 1. Introduction

The coronavirus disease (COVID-19) was declared a global pandemic in March 2020, which has since received rapid response with research on physical treatments and vaccines. An increasing body of evidence suggests that many patients infected with COVID-19 have developed neurological symptoms [1]. Studies concerning mental health consequences of the pandemic and associated lockdown restrictions are of no less importance.

The prevalence of anxiety and depression symptoms, as well as the risk of substance use, suicides, and domestic violence, has increased throughout the pandemic. Anxiety and fear were seen particularly with panic buying behaviours across the world [2,3].The COVID-19-driven hoarding and panic buying behaviour by the mass population left supermarket shelves in many locations empty for weeks. Panic buying is defined as when consumers purchase an exceptionally large amount of products in the anticipation of a shortage. We have not seen panic buying behaviour to this extent in previous disease outbreaks [4].

The hoarding of a particular household item, —toilet paper—was observed in several countries, including the U.S., Australia, Canada, the U.K., Singapore, Japan, Hong Kong and other cities [5]. This widespread phenomenon received a lot of public attention in the news and social media, sending shockwaves to demand and supply chains around the globe [6]. The toilet paper shortage has led to several cases of people fighting in stores. For example, two women in Australia were charged for a physical altercation as a result of arguments over toilet paper buying [7].

There are several theories for the panic buying phenomenon. One potentially rational reason for panic buying is to minimise the risks of running out of supplies during emergencies and pandemics. Panic buying, either rationally or irrationally, has been a natural human response to imminent disasters and emergency crises in an attempt to protect and regain control of fear [8]. However, repeated exposure to media coverage on the pandemic could result in psychological distress [9]. Prolonged exposure to traumatic events as such is a known risk factor linked to adverse physical health and mental health even in the absence of post-traumatic stress disorder (PTSD) [10]. Stockpiling daily necessities may offer comfort over unpredictable circumstances [11]. The expectation of stock scarcity or the surge in pricing may lead to panic buying behaviours resulting from fear and anxiety. Herding mentality could also account for the panic buying behaviour, as people could be influenced by others to make irrational emotional decisions [12].

Social media was thought to be amplifying the sense of supply scarcity, with stories about people engaging in hoarding supplies being highlighted in the media. The use of Twitter, a social media platform, has become increasingly widespread across populations as a way to engage and participate in civic life [13]. It allows users to reflect their emotions and express their opinion on given topics. Twitter provides data on public conversations that has demonstrated utility for academic research on current public health issues [14], e.g., on harmful alcohol use of young people [15], to monitor mental health discussions [16], and infodemiology studies of the H1N1 pandemic [17,18].

In this study, we aim to examine the contents on social media surrounding the anxiety-induced toilet paper panic buying behaviour that occurred during the COVID-19 pandemic. Analyses on social media information often adopt qualitative research methodologies by applying a grounded theory or phenomenological approach. The grounded theory approach emphasises constructing theories based on empirical data, instead of testing a priori hypotheses [19]. This enables the generalisation of theories through an in-depth comprehension of the concurrent phenomenon using social media contents. Making use of social media contents provides an opportunity to examine human behaviours, mentality, and conversations in the era of digital communications. Knowledge gained from social media data could guide strategies to address public concerns and reduce anxiety-induced panic behaviours in future crises. This study draws on public conversations from Twitter’s open-source database to examine how anxiety-related panic buying behaviour in the time of COVID-19 is affected by public concerns, comments, opinions and attitudes on Twitter related to toilet paper hoarding during the pandemic.

## 2. Methods

### 2.1. Qualitative Approach

This study follows the Standards for Reporting Qualitative Research (SRQR) guidelines (see Table A1 (Appendix A)). We applied Glaser’s grounded theory approach to develop themes and theories from the dataset using an interpretivist paradigm [20]. Theories behind the panic buying phenomenon were constructed through methodical gathering and analysis of collected tweets by deriving keywords and subthemes, then coding them into relevant sub-themes and themes before developing into theories. This approach could formulate hypotheses based on existing empirical tweets, which could assimilate the data by continually reviewing the sub-themes and themes developed throughout the coding process. New themes apart from existing ones in the literature could be generated. In addition, the interpretivist paradigm stresses the interpretation of information through data collection and methodological analysis, to explain subjective reasons behind phenomenological behaviours [21].

### 2.2. Researcher Characteristics

The researchers that carried out this research are based in Australia, aged 25–35, with backgrounds in psychology, epidemiology, and statistics. The researchers are not, and do not have acquaintances, who are prolific social media users.

### 2.3. Context

Our research focused on tweets that were in English. From the pilot data downloaded, we found that about half of the tweets were in English (~50%; see Figure A1. Tweets in Japanese (~14%), Spanish (~10%) and French (6%) were also common. However, the researchers are not proficient in languages other than English, therefore only tweets in English were included.

### 2.4. Sampling Strategy

Using our search terms, we retrieved 255,171 unique tweets. 251,090 tweets that had less than 10 retweets were excluded. The reason was to select a sample of tweets that had a higher level of exposure and had been shared by multiple users to represent the mass conversation. Based on our inclusion criteria (see Figure A2), 4081 tweets were deemed eligible to be included in the present study (see Figure A2).

Of these, 100 tweets were first randomly selected to develop a coding scheme in the initial coding phase. In the focused coding phase, tweets were qualitatively coded in batches of 100 random tweets from the remaining dataset using the coding scheme developed from the initial coding phase. The tweets were then coded based on a data saturation approach [22], a sampling method that estimates the likely number of qualitative data needed to reach saturation for a given study [22]. The process was repeated (by taking another 100 tweets) and coded for its themes until no new themes were observed in the batch. That is, when all of the 100 consecutive tweets analysed in a given batch conveyed the same themes as the previous batch of 100 tweets. We used this approach to determine the optimal number of tweets required to generate summaries that are representative of the overall sample.

### 2.5. Ethical Issues Pertaining to Human Subjects

This is an observational study of publicly available data from Twitter, an open exchange platform that provides access to public conversation data for academic research. Our study was conducted in ethical accordance with Twitter’s rules around data usage for non-commercial academic research. We used publicly available tweets and our study did not involve tweeting, retweeting or liking contents. Usernames were removed from the datasets before analysis. Our aims, analysis, and summarization methods were reviewed and approved by Twitter (ref:00DAOKOA8).

### 2.6. Data Collection

Original tweets about COVID-19 related hoarding of toilet paper were collected between 29 February and 29 March 2020. The search terms included “toilet paper”, “toilet roll”, #toiletpaper, #toiletpapercrisis, #ToiletPaperEmergency, #toiletpapergate, #toiletpaperwars, #toiletpaperroll, and #toiletpaperrolls.

### 2.7. Data Processing

Tweets were collected into a csv file, merged, and duplicates were removed. Only the text contents of tweets were collected (i.e., users’ information was not collected).

### 2.8. Data Analysis

We analysed the content of tweets using an approach consistent with studies of opinions in previous infectious disease outbreaks, e.g., [17]. The analysis was carried out in four steps involving the use of iterative process (see Figure 1): (i) researchers conducted data collection and screening of tweets; (ii) two researchers (CT and JC) reviewed tweets in the initial coding phase to develop the preliminary coding scheme for focused coding. Keywords for each subtheme and sentiment were identified (see Table A2); (iii) in the focused coding phase, tweets were randomly cut into batches of 100 tweets for each coding session. Sentiments of the tweets were analysed and coded for each of the tweets by analysing their content and categorised into positive, neutral or negative (see Table A2). After each coding session, themes and sub-themes were developed through revisions and amendments to assimilate the data. The coders then discussed and agreed upon the sub-themes before developing the themes for thematic analysis. Key themes were then created based on the sub-themes. Disagreement of themes and coding were resolved at the end of each coding sessions. Coding was terminated when data saturation was met as no new themes or sub-categories emerged from two consecutive coding sessions; (iv) theories were then generated from the themes and identified and existing literature to explain the findings.

### 2.9. Intercoder Reliability

Measures of reliability were calculated for each individual code, comparing the agreement between both coders. Overall, there was a percentage agreement of 87% with a Cohen’s Kappa coefficient of 0.42.

## 3. Results

### 3.1. Tweet Characteristics

Of all the Tweets analysed until saturation reached at *n* = 500 (see Figure A2), 46% carried negative sentiments regarding the panic buying phenomena of toilet paper, 13% carried positive sentiments, with the remaining 31% having neutral sentiments (see Figure 2). Tweets were 3.5 times more likely to carry negative sentiments than positive sentiments.

Distributions of retweets and likes of the toilet paper-related tweets are presented in Figure 3. The majority (87%) of tweets had 10 to 99 retweets (Table 1). The five most shared tweets had been retweeted more than 1000 times. In addition, 88% of tweets received 0 to 499 likes, and 6% of tweets received more than 1000 likes.

### 3.2. Influential Tweets

Among the most influential tweets (Table 2), half of them contained negative sentiments, with expressions of frustration or anger over the lack of toilet rolls due to panic buying, or sarcasm, humour, and confusion towards the phenomenon. Some of the highly influential tweets were supportive and provided up-to-date information about the panic buying situation.

### 3.3. Themes

Five prominent themes were identified from the data, which include: (1) Humour or sarcasm; (2) Opinion and emotion; (3) Personal experience; (4) Support or information; and (5) Monetary (see Table 3).

#### 3.3.1. Theme 1: Humour and Sarcasm

Humour and sarcasm were the most commonly expressed themes of the toilet paper-related tweets (26%). These tweets were attempting to be amusing or eliciting humour either sarcastically or generally as a joke regarding the toilet paper panic buying situation. This included specific humour related to using alternatives as toilet paper like newspaper and cabbage leaves, mocking the insanity of panic buying using sarcastic metaphors, making jokes about how the future generations would view this frantic pandemic or making political references. For example:


*Even in a video game toilet paper is limited…FML #AnimalCrossing #ACNH #NintendoSwitch*



*Since toilet paper now available at our nearby supermarket, we are cancelling our newspaper subscription.*


#### 3.3.2. Theme 2: Opinion and Emotion

Many of the tweets carried strong opinions regarding the situation. They contained views and beliefs from the users which carried their feelings and emotions. Anger was the most dominant emotion (24%), followed by having sceptical (13%) or unconcerned (6%) opinions. Twitter users expressed feeling angry and frustration towards the insufficiency of toilet paper and the frantic panic buying phenomenon. Some other users felt sceptical about the necessity of stockpiling toilet paper as they were more rational about the situation. Some were unconcerned about the current panic buying of toilet paper and often they applied a humour or sarcastic approach of expressing it. For example:


*Does the #coronavirus make you sh*t yourself to death then? Why the hell is everyone stockpiling #toiletpaper?!*


#### 3.3.3. Theme 3: Personal Experience

This theme included tweets about events that individuals experienced, negatively or positively. Some of these tweets referred to negative personal experience of failing to obtain toilet paper due to insufficient supply (10%) and unpleasant shopping experience (6%). Only a few expressed joys over obtaining adequate toilet paper (3%) and selfless acts of helping others (1%). Specifically, people posted about negative experiences in obtaining toilet paper or running low on toilet paper without being able to purchase more from the shops. Others expressed positive experience as they had access to toilet paper or were helping others with obtaining toilet paper. For example:


*I only have six spare toilet rolls and one in use. Living life on the edge! #panicbuyinguk.*


#### 3.3.4. Theme 4: Support or Information

This theme included tweets providing support and information regarding panic buying of toilet rolls. They included tweets of shared information relating to the panic buying situation such as where to obtain toilet paper (15%), news from media (7%) and providing support (7%). Three sub-themes of support, information and media were identified in the coding process. Some Twitter users shared information such as website links and articles regarding where to obtain toilet paper or the current extent of panic buying. Other users shared news regarding the panic buying phenomenon whereas some provided support or expressed they have received support from others for obtaining toilet rolls. For example:


*Women fight over toilet paper at Australia supermarket amid #coronavirus fears. https://www.ndtv.com/world-news/watch-women-fight-over-toilet-paper-at-australia-supermarket-amid-coronavirus-fears-2191709.*


#### 3.3.5. Theme 5: Monetary

This theme included tweets that are related to money such as profiteering or promoting a business. Some were trying to use the situation to promote their business or their Twitter account through offering discounts for toilet paper or actual toilet rolls for their followers and new followers. Others were advertising and profiteering for oneself through stockpiling toilet paper, and selling them at a higher market price on Twitter. For example:


*SHINY GIVEAWAY Retweet and Follow for a chance to win a full pack of shiny, Scott toilet paper rolls. Your local market ran out of them, but I haven’t. Will pick 5 winners at the end of the week! Yup giveaways are back!!*


## 4. Discussion

The coronavirus 2019 pandemic has led to several unprecedented events in modern history, including the strictly enforced social distancing rules to prevent contagion, the tremendous disruption to countless businesses, and the panic buying phenomena observed across nations. Several plausible theories could provide explanations to the irrational hoarding behaviours. From the previous global crisis situations, panic buying behaviours could act as a coping mechanism to control fear and anxiety. While fear is influenced by the perception of actual and perceived threat, the response to fear could be driven by emotions to maintain control over the ambivalence. Hoarding of food and necessities could offer temporary consolation to relieve anxiety and regain a certain degree of control, which often occurs in pandemics and economic crises [11]. During the 2003 SARS outbreak, panic buying of salt, rice, vinegar, and other perishable goods occurred in certain provinces of China and in Hong Kong [23]. However, the scale of panic buying during the COVID-19 pandemic has been unparalleled. The uncertainties about the duration of the pandemic and how it would affect our lives inflict profound psychological impacts on people [24]. Our study of tweets regarding panic buying of toilet paper by in vivo analysis examines the psychology behind the behaviour.

From our analysis, close to half of the tweets expressed negative feelings towards the panic buying of toilet paper, such as anger and frustration over the insufficient stock of toilet rolls from the stores. The top three most influential tweets had a total of 588,000 likes and 80,000 retweets, all of which were negative, expressing either confusion or anger towards the panic buying phenomena. We theorised that the popularity and influence of negative tweets could potentially lead to a “snowball effect” on social media, as regular social media users pick up sentimental feelings and opinions from tweets by others [13]. This is enhanced by the belief that they would consume more toilet paper than normal due to the lockdown and work-from-home restrictions. In addition, unlike canned food, pasta or masks and hand sanitisers, toilet paper is not an essential necessity for the pandemic. However, empty shelves of toilet paper are more noticeable in supermarkets, as toilet papers are bulkier and take up more space than other perishable items. The exposure to these photos on social media acts as an emotional trigger of anxiety and panic of running out of toilet paper. Social media users are more likely to assimilate fearful emotions from these tweets, which could fuel anxiety-driven panic buying behaviours.

Social media may have reinforced the feeling of helplessness, such as the lack of necessities, which could lead to emotional distress and depression. During the H1N1 outbreak in 2009, Twitter analysis studies have found that misinformation and confusion regarding the use of informal terms could be easily spread through social media [17]. However, positive tweets were also found in this study. In our study, results indicated that a quarter of the tweets contain supportive information regarding supplies, discounts of toilet paper and prompting others to stay calm in midst of the crisis. Some retailers and manufacturers had made tweets empathising the sufficiency of toilet paper supplies, in hope to counterbalance the spread of anxiety and panic buying behaviours. Using digital platforms to enhance access to health services including mental health support is not a new idea, however, general uptake of digital mental health applications has been low in many countries [25]. The COVID-19 pandemic may be a good opportunity to explore the feasibility of digital health options [26].

Our findings provided useful data for the understanding of social psychology by comprehending public sentiments and opinions. This could equip nations around the world to be better prepared for another emergency crisis. Similar to crises during previous economic depressions, efforts from state governments must be deployed to minimise underlying rational and irrational incentives of anxiety-induced panic buying. While it is crucial for manufacturers and retailers to maintain a steady supply and production, public broadcast and communication could effectively minimise the uncertainty and fear among the public. The government could utilise the power of social media, as to monitor sentiments and efficiently identify and address public concerns by issuing pertinent state-wise announcements, especially to tackle mass panic and anxiety. It is evident that clear and early messages need to be communicated to the public to prevent panic behaviours, for example, posts from manufacturers of details on their remaining stock in storage and the estimated time for restocking, while limiting purchase per customer as early as possible. This could efficiently easepublic concerns of running out of daily necessities.

### Limitations

Our analysis focused on text-based tweets and did not analyse tweets that were of pictures, pictures of texts, or videos. This is a limitation because many people go to Twitter for viewing news, photos, and videos. For example, an excluded tweet entitled “The math on toilet paper” had over 5000 retweets but was excluded because it was a video tweet. The video showed calculations to explain that a person who purchased four cases of toilet paper for a family of four, which were required to quarantine for 14 days would need to go to the toilet 182 times a day to use the purchased amount, calling the public to calm down. We lost opportunities for analysis due to our exclusion of media tweets, but a hand search and coding of a sample of media-based tweets reviewed that they were consistent with the themes identified and contents of our results. Review and analyses of media attached to each tweet were outside of our research team’s capacity due to manual retrieval and coding requirements. With the advancement in information technology, future research can more efficiently analyse attitudes, emotions and themes based on images and videos. For example, Google Cloud’s Vision API have pre-trained machine learning models that can detect text within images, and emotions (e.g., joy, anger, surprise) from images, then code them into predefined categories. Perhaps future studies could make good use of artificial intelligence software to conduct these analyses.

The inferentiality of the tweets does not represent the totality across time. Tweets posted earlier may have more time to get retweeted and liked initially, but then generally lose further interaction after 3–5 days. Therefore, to reduce this bias, we collected the data with a lag time of a week after our target research period. Our supplementary analyses confirmed that there were no significant correlations between tweet date and number of retweets (r = 0.01, *p* = 0.537) or likes (r = 0.01, *p* = *0*.631; see Figure A3). On the other hand, intercoder reliability of this study is at a moderate level, this might be due to the exploratory nature of this research, where themes were constantly reviewed and disagreements were resolved at the end of each coding session until saturation was reached.

The use of hashtags is not consistent across different populations. Certain cohorts and socio-demographic groups may differ in the use of hashtags in their posts. Therefore, to increase the generalisability of our findings, we included both ‘toilet paper’ and ‘toilet roll’ as search terms, rather than searching for hashtags only. Still, findings could not be generalised across everyone on Earth because not everyone uses Twitter, and among those who use, some are read-only users. In addition, 20% of Twitter’s daily users are from the USA, and in general are younger and have higher socio-economic status [27]. Nevertheless, due to the large number of 145 million daily active users and 500 million posts per day, our findings represent conversations from a substantial number of people within the society. Finally, the use of some trending terms related to the toilet paper hoarding incidence such as “toiletpapercrisis”, “toiletpaperemergency”, “toiletpaperwars” may have led to the significant proportion of tweets carrying negative sentiment. In Australia, for example, hoarding of toilet paper and other essential supplies has been widely portrayed negatively in the media, and condemned publicly by the Prime Minister. Therefore, tweets agreeing the condemned behaviour would not be a majority viewpoint. Nonetheless, the limitations should be recognised when interpreting the results.

## 5. Conclusions

In the modern epoch of technology, social media has become one of the most dominant tools of information dissemination and can be drawn upon to gain insights on population mental health. During the COVID-19 pandemic, anxiety-induced behaviours could be observed by the vast amount of posts regarding toilet paper buying shared on social media platforms such as Twitter. Our study found that while many people on Twitter expressed negative sentiments and frustration towards the panic buying of toilet paper, some tweets providing supportive and positive information were quite influential among users as evident by the number of retweets and likes. Twitter data provides an opportunity for research into public sentiments, conversations, and concerns at a given time. It would be worthwhile to further invest into social media data analysis for research on public mental health.

## Figures and Tables

**Figure 1 ijerph-18-01127-f001:**
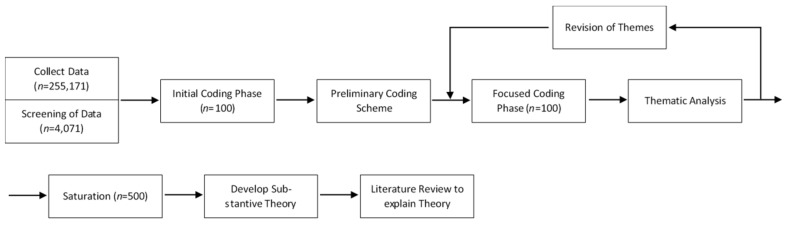
Flowchart of data analysis process.

**Figure 2 ijerph-18-01127-f002:**
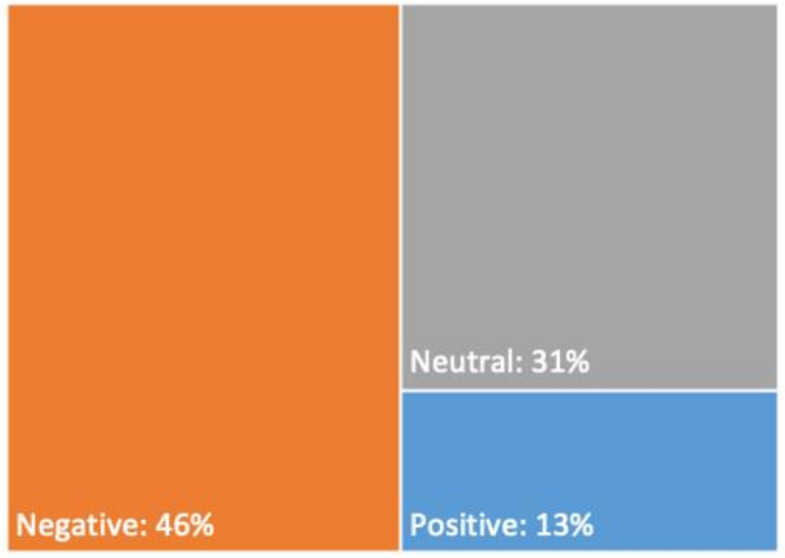
Sentiments of toilet paper-related tweets.

**Figure 3 ijerph-18-01127-f003:**
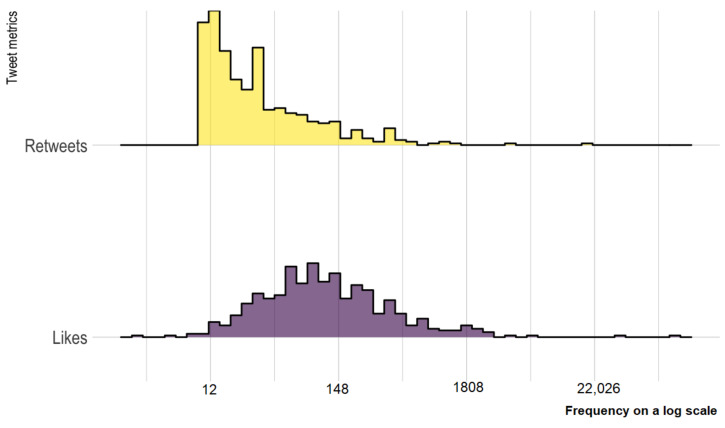
Distribution of retweets and likes from included toilet paper-related tweets.

**Table 1 ijerph-18-01127-t001:** Descriptive statistics of retweets and likes of the original tweets.

	Frequency of Original Tweets		Totality of Engagement	
	*n*	%	Σ	%
**Retweets**				
Total	500	100%	53,141	100%
By number of Retweets				
10–99 RTs	433	86.6%	11,679	24.7%
100–999 RTs	62	12.4%	15,371	34.2%
1000–1999 RTs	3	0.6%	3774	8.4%
2000–4999 RTs	1	0.2%	3895	9.0%
5000+ RTs	1	0.2%	18,422	23.8%
**Likes**				
Total	500	100%	272,630	100%
By number of likes				
0–499	422	88.4%	55,710	20.4%
500–999	31	6.2%	21,545	7.9%
1000–4999	24	4.8%	47,728	17.5%
5000–9999 likes	1	0.2%	6216	2.3%
10,000–29,999 likes	0	0%	0	0%
30,000+ likes	2	0.4%	141,431	51.9%

**Table 2 ijerph-18-01127-t002:** Most influential tweets based on retweets and likes.

Likes		Retweets		Theme	Tweet
Rank	Number	Rank	Number		
1	424,694	1	55,378	Humour, Sceptical, Negative	The year is 2024. Coronavirus was worse than we expected. Toilet paper has been gone for 3 yrs. I have over 250 rolls from not wiping my a** for yrs. The police are approaching. They have surrounded my home. My gun is loaded aiming at the door. They want my paper. This is the end
2	108,571	2	18,422	Anger, Information Sharing, Insufficient Supply, Negative	At my local supermarket, a clerk told me that their fresh shipment of toilet paper sold out in 15 min. People preparing for the sh*tpocalypse.
3	55,430	3	6149	Humour, Anger, Negative	Shoutout to all the a****** who took an unnecessary amount of toilet paper--now elderly people and families who need it are being punished for your greed and selfishness.
4	32,860	5	3895	Humour, Anger, Negative	Y’all gonna feel dumb as fuck in 2 weeks when everything is back to normal but u just got 158 rolls of toilet paper stored in your linen closet
5	32,108	4	5514	Sceptical, Media-related, Negative	can someone explain to me the need for 24 rolls of toilet paper. is it because the media is full of crap?
6	25,913	--	982	Anger, Unpleasant Shopping, Negative	Just been in the asda and this man had aba 80 toilet rolls at the check out and the women on the till was like ‘bloody hell how much toiletting you planning on doing’ *n* he flipped *n* was like what the f**ks it got to do with you ye cheeky b**ch
7	24,770	--	1455	Support, Neutral	everyone out there buying toilet paper but make sure you guys have enough pet food okay
8	17,262	--	934	Humour, Neutral	when ppl in new york say “I’m stocking up on toilet paper’ all I hear is “I have bathroom storage, like a duchess or an earl”
9	17,139	--	3118	Humour, Sceptical, Neutral	STAGE 1 (cont’d): I’m safe, everybody is overreacting, what’s the need to go out with masks and stock toilet paper? I’m going to live my life as usual, there’s no need to freak out. STAGE 2. The number of cases begins to be significant.
10	16,827	--	2060	Anger, Support, Neutral	I feel so bad for cashiers at grocery stores right now… ×10 busier than Christmas, dealing with adult children fighting over toilet paper, having to come in contact with hundreds of people, touching dirty money. Be extra nice to retail workers :(
--	10,908	6	3632	Support, Neutral	The most vulnerable populations often can’t get even basic staples. If there is a high risk individual you know (elderly or infirm), see if you can help. Maybe you don’t need fifty rolls of toilet paper, but they might need a few. Be proactive. Don’t worry. #CareForOthers.
--	8888	7	3626	Monetary, Information Sharing, Neutral	GIVEAWAY-20 rolls of toilet paper! (RT) & (Like) to win!
--	17,139	8	3118	Humour, Sceptical, Neutral	STAGE 1 (cont’d): I’m safe, everybody is overreacting, what’s the need to go out with masks and stock toilet paper? I’m going to live my life as usual, there’s no need to freak out. STAGE 2. The number of cases begins to be significant.
--	14,287	9	2955	Support, Positive	God bless the American manufacturers who are making and packaging food, cleaning supplies, and other essential items (including toilet paper). We are #InItTogether!
--	16,632	10	2936	Humour, Neutral	Early Spring, 2050 Your Millennial grandmother sends you another case of 72 rolls of toilet paper “just in case.” You understand it’s a thing with that whole generation, but as you stack all of it in the linen closet once again you wish you could convince her to stop.

-- Not ranked within the top ten most influential Tweet.

**Table 3 ijerph-18-01127-t003:** Themes, with counts, and stats of sentiment.

Themes and Sub-Themes Identified	Definition	Percentage
**(1) Humour or sarcasm**	**Aimed to be amusing, comical, or to illicit humour**	**26%**
**(2) Opinion and emotions**	**Carried emotions, views and beliefs**	
Anger	Anger and frustration over others’ panic buying behaviour	24%
Sceptical	Scepticism over the need to panic buy	13%
Unconcerned	Tweets that express unconcern over panic buying	6%
**(3) Personal experience**	**Tweets about events that individuals experienced**	
Negative		
Insufficient supply	Inability to obtain toilet paper	10%
Unpleasant shopping experience	Negative experience of shopping for toilet paper	6%
Positive		
Access to toilet paper	Able to gain access to toilet paper	3%
Altruism	Tweets related to people helping others regarding toilet paper	1%
**(4) Support or information**	**Tweets that provided support or information**	
Information	Tweets that provided information	15%
Media	Tweets by news outlets	7%
Support	Tweets about providing or receiving support	7%
**(5) Monetary**	**Money-related, such as trying to promote a product or profiteering through toilet paper**	**4%**

*Note.* Sum of percentages exceed 100% because some tweets belonged in more than one theme.

## Data Availability

The datasets generated during and/or analysed during the current study are available in the ResearchGate repository, https://doi.org/10.13140/RG.2.2.33547.03362.

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
