# Peer review of "Anxiety and Panic Buying Behaviour during COVID-19 Pandemic—A Qualitative Analysis of Toilet Paper Hoarding Contents on Twitter"

_ijerph, 2021, doi:10.3390/ijerph18031127_

Round 1

Reviewer 1 Report

Abstract: the number of results examined is misleading. As I understood, from reading the rest of the article, only 500 results were actually analysed qualitatively. Also given the title of the article “qualitative analysis” I expected a greater focus on this method in the abstract rather than the number of results analysed.

Methods: Qualitative approach. The authors state that they use SRQR but they do not follow all the recommended guidelines. Notably missing “several qualitative reporting criteria recommend that authors describe how they implemented a presumably familiar technique in their study rather than simply mentioning the technique. For example, authors often state that data collection occurred until saturation, with no mention of how they defined and recognized saturation. Similarly, authors often mention an “iterative process,” with minimal description of the nature of the iterations. The SRQR emphasizes the importance of explaining and elaborating on these important processes.” (https://pubmed.ncbi.nlm.nih.gov/24979285/).

Methods: Qualitative approach. There are different approaches to grounded theory. Which approach to grounded theory was used in this study?

Methods: Sampling strategy. I find the arguments for the sampling strategy confusing. On the one hand the authors argue that they are concentrating on tweets that have mass exposure (10 or more retweets) but on the other they are argue for a grounded theory approach. If saturation is important then surely all tweets should be considered, not just mass exposure tweets? And indeed in the Ahmed et al article cited by the authors, going beyond popular tweets is considered important.

Methods: Sampling strategy. The explanation for how saturation was reached could be clearer (see earlier comment). It would be helpful to state here how many tweets in total were selected.

Methods: Data analysis. How the authors taught themselves to analyse the data should not be described as a step (“researchers conducted literature review on Twitter analysis methodologies ….”). The authors should be describing how the data was actually analysed.

Methods: Data analysis. How was it decided whether tweets were positive, neutral or negative? A codebook should be included here.

Methods: Data analysis. Was intercoder reliability assessed? If so, how was this done and how similar was the application of codes?

Methods: Data analysis. What evidence is there that data saturation is met when 20% of consecutive tweets have same tweets? Perhaps it is the way this has been phrased, but it doesn’t sound like saturation has been met.

Ethics. This has not been addressed and must be.

Results: Tweet characteristics. Given the data collection terms (toiletpapercrisis, toiletpaperemergency, toiletpaperwars) it is not surprising that 46% of the tweets carried negative sentiment but the authors do not address or discuss this.

Results: Figure 2 appears to be missing? Presumably an upload error.

Results. What data is analysed in Figure 3? Is this the whole dataset of 255k tweets? If so, why the authors sometimes use the whole dataset and at other times a subset needs to be justified / clarified.

Discussion: Although there is plenty of scope for novelty (and a more in-depth analysis of the toilet paper tweets would be good), as the paper is currently written, it is not clear what the novelty of this research.

  1. Using twitter to monitor sentiment is not a new idea. See for example Cano, Amparo E., Suvodeep Mazumdar, and Fabio Ciravegna. "Social influence analysis in microblogging platforms–a topic-sensitive based approach." Semantic Web5 (2014): 357-372.
  2. There is a lot of recent work analysing twitter in relation to Covid-19. Ahmed, W., Vidal-Alaball, J., Downing, J., & Seguí, F. L. (2020). COVID-19 and the 5G conspiracy theory: social network analysis of Twitter data. Journal of Medical Internet Research22(5), e19458. McKendry, R. A., Rees, G., Cox, I. J., Johnson, A., Edelstein, M., Eland, A., ... & Heymann, D. (2020). Share mobile and social-media data to curb COVID-19. Nature580(7801), 29.
  3. It is also not clear what this methodological approach adds. The authors claim to follow SRQS which suggests that “The purpose of qualitative research is to understand the perspectives/experiences of individuals or groups and the contexts in which these perspectives or experiences are situated”. This paper does not do this.

Discussion: no theoretical framework is deployed in this paper. The researchers have a background in psychology and epidemiology, and could add some really interesting insights to this paper. I would like to see a much more in-depth analysis / discussion with insights / interpretations that are rooted in psychology / epidemiology.

Author Response

Response to Reviewer 1’s Comments

Reviewer 1

1.1        Abstract: the number of results examined is misleading. As I understood, from reading the rest of the article, only 500 results were actually analysed qualitatively. Also given the title of the article “qualitative analysis” I expected a greater focus on this method in the abstract rather than the number of results analysed.

>> Response: We have now revised the abstract to clarify the method.

“Methods: A total of 255,171 Tweets were collected. Of these 4,081 met our inclusion criteria and 100 Tweets were randomly selected to develop a coding scheme with keywords, sub-themes and sentiment definitions in the initial coding phase. 500 Tweets were then qualitatively analysed in the focused coding phase until saturation was met. Key themes were then identified before theories were generated.”

1.2        Methods: Qualitative approach. The authors state that they use SRQR but they do not follow all the recommended guidelines. Notably missing “several qualitative reporting criteria recommend that authors describe how they implemented a presumably familiar technique in their study rather than simply mentioning the technique. For example, authors often state that data collection occurred until saturation, with no mention of how they defined and recognized saturation. Similarly, authors often mention an “iterative process,” with minimal description of the nature of the iterations. The SRQR emphasizes the importance of explaining and elaborating on these important processes.” (https://pubmed.ncbi.nlm.nih.gov/24979285/).

>> Response: Our apologies, we have checked to make sure that we have not missed any items in the SRQR guidelines. This guidelines is now being included as a new appendix table A. A flowchart has now been added to demonstrate the use of iterative process in our research.

“This study follows the Standards for Reporting Qualitative Research (SRQR) guidelines (see Appendix A).”

“The analysis was carried out in four steps involving the use of iterative process (see Figure A) “

1.3        Methods: Qualitative approach. There are different approaches to grounded theory. Which approach to grounded theory was used in this study?

>> Response: We undertook a Glaserian classic grounded theory approach to grounded theory (Glaser, 1978). We have now revised our paper to better specify the approach.

“We applied Glaser’s grounded theory approach to develop themes and theories from the dataset using an interpretivist paradigm (Charmaz & Belgrave, 2007)”

1.4        Methods: Sampling strategy. I find the arguments for the sampling strategy confusing. On the one hand the authors argue that they are concentrating on tweets that have mass exposure (10 or more retweets) but on the other they are argue for a grounded theory approach. If saturation is important then surely all tweets should be considered, not just mass exposure tweets? And indeed in the Ahmed et al article cited by the authors, going beyond popular tweets is considered important.

>> Response: Our decision to concentrate analysis on Tweets that had a minimum exposure of 10 retweets developed through our grounded theory approach. In addition, while it may be desirable to include every case, it was not plausible to analyse all of the 255,171 Tweets due to our limited capacity, therefore we decided to prioritize Tweets with at least 10 retweets. Using theoretical sampling to target Tweets with at least 10 retweets, we aimed to reduce noise in the sample and capture a representative sample of individuals who were directly addressing the toilet paper phenomenon. We agree that Ahmed et al. highlight the importance of investigating all levels of Tweets. This paper also highlights the importance of ‘broadcasting hubs’ that act as central nodes for the distribution of information. We analysed all levels of Tweet popularity through randomisation while also conducting separate analysis on the most retweeted Tweets which we believe represent the central nodes of information discussed in Ahmed et al. We believe this sampling strategy captured the full spectrum of Tweet popularity.

1.5        Methods: Sampling strategy. The explanation for how saturation was reached could be clearer (see earlier comment). It would be helpful to state here how many tweets in total were selected.

>> Response: We have revised our sampling strategy to better explain how saturation was reached, as follows:

“We adopted a data saturation approach, a sampling method that estimates the likely number of qualitative data needed to reach saturation for a given study (Guest, Namey, & Chen, 2020). The process was repeated (by taking another 100 Tweets) and coded for its themes until no new themes were observed in the data when 20% of consecutive Tweets conveyed the same themes. We used this approach to determine the optimal number of Tweets required to generate summaries that are representative of the overall sample.”

1.6        Methods: Data analysis. How the authors taught themselves to analyse the data should not be described as a step (“researchers conducted literature review on Twitter analysis methodologies ….”). The authors should be describing how the data was actually analysed.

>> Response: Apologies, we have now fixed this.

“Analysis process was carried out in four steps: i) researchers conducted data collection and screening of Tweets.”

1.7        Methods: Data analysis. How was it decided whether tweets were positive, neutral or negative? A codebook should be included here.

>> Response: Thank you for the comment, we have revised our data analysis methods section

“Sentiments of the Tweets were analysed and coded for each of the Tweets by analysing their content and categorising into positive, neutral or negative (see Appendix D).”

1.8        Methods: Data analysis. Was intercoder reliability assessed? If so, how was this done and how similar was the application of codes?

>> Response: Inter-rater reliability was assessed by calculation of percentage agreement and a kappa coefficient. These measures of reliability were calculated for each individual code, comparing the agreement between both coders. Overall there was an agreement of 87% with a kappa of 0.42. We believe these diminished levels of inter-rater reliability may be due to the exploratory nature of this study.

“Intercoder reliability 

Measures of reliability were calculated for each individual code, comparing the agreement between both coders. Overall, there was a percentage agreement of 87% with a Cohen's Kappa coefficient of 0.42.”

“On the other hand, intercoder reliability of this study is at a moderate level, this might be due to the exploratory nature of this research, where themes were constantly reviewed and disagreements were resolved at the end of each coding session until saturation was reached.”

1.9        Methods: Data analysis. What evidence is there that data saturation is met when 20% of consecutive tweets have same tweets? Perhaps it is the way this has been phrased, but it doesn’t sound like saturation has been met.

>> Response: We apologise for the awkward wording. The 20% referred to the last batch of n=100 Tweets analysed before saturation was met at n=500. When we were no longer extracting novel themes from a given batch of 100 Tweets, we considered saturation to be achieved. We have now revised our methods to clarify the process.

“The process was repeated (by taking another 100 Tweets) and coded for its themes until no new themes were observed in the batch. That is, when all of the 100 consecutive Tweets analysed in a given batch conveyed the same themes as the previous batch of 100 Tweets.”

1.10      Ethics. This has not been addressed and must be.

>> Response: We have now addressed this, as follows,

“Our study was conducted in ethical accordance with Tweeter’s rules around data usage for non-commercial academic research. We used publicly available Tweets and our study did not involve Tweeting, Retweeting, or liking content. Usernames were removed from the datasets before analysis. Our aims, analysis, and summarization methods were reviewed and approved by Twitter (ref:00DAOKOA8).”

1.11      Results: Tweet characteristics. Given the data collection terms (toiletpapercrisis, toiletpaperemergency, toiletpaperwars) it is not surprising that 46% of the tweets carried negative sentiment but the authors do not address or discuss this.

>> Response: Thank you for your comment. The authors agree that this is a limitation of our data collection terms and it has been discussed in the last paragraph of the Discussion.

“Finally, the use of some trending terms related to the toilet paper hoarding incidence such as “toiletpapercrisis”, “toiletpaperemergency”, “toiletpaperwars” may have led to the significant proportion of tweets carrying negative sentiment. In Australia, for example, hoarding of toilet paper and other essential supplies has been widely portrayed negatively in media, and condemned publicly by the Prime Minister. Therefore, tweets agreeing the behaviour would not be a majority viewpoint. Nonetheless, the limitation should be recognised when interpreting the results.”

1.12      Results: Figure 2 appears to be missing? Presumably an upload error.

>> Response: Thank you. This is now fixed.

1.13      Results. What data is analysed in Figure 3? Is this the whole dataset of 255k tweets? If so, why the authors sometimes use the whole dataset and at other times a subset needs to be justified / clarified.

 >> Response: The data analysed in Figure 3 is the whole dataset of 255k tweets, this was analysed to justify the reason why we excluded Tweets with less than 10 Retweets, which was to select a sample of Tweets that had a higher level of exposure and had been shared by multiple users to represent the mass conversation.

1.14      Discussion: Although there is plenty of scope for novelty (and a more in-depth analysis of the toilet paper tweets would be good), as the paper is currently written, it is not clear what the novelty of this research.

  1. Using twitter to monitor sentiment is not a new idea. See for example Cano, Amparo E., Suvodeep Mazumdar, and Fabio Ciravegna. "Social influence analysis in microblogging platforms–a topic-sensitive based approach." Semantic Web5 (2014): 357-372.

 >> Response: The authors agreed that there is existing literature on using Twitter to monitor sentiment. Our research demonstrated the role of Twitter on inducing mass-anxiety on the insufficiency of toilet paper. It is interesting that toilet paper, unlike food, water or medical supplies, is not an essential necessity to survive the pandemic, However, we have seen mass panic buying of toilet paper around the globe. Our findings indicated that Tweets were 3.5 times more likely to carry negative sentiments than positive sentiments, where the top 5 influential Tweets with a total of over 650,000 likes carried negative sentiments. We theorised that the instantaneousness of Tweets along with sentimental contexts, fuelled fear and panic among social media users, triggering panic buying behaviours.

  1. There is a lot of recent work analysing twitter in relation to Covid-19. Ahmed, W., Vidal-Alaball, J., Downing, J., & Seguí, F. L. (2020). COVID-19 and the 5G conspiracy theory: social network analysis of Twitter data. Journal of Medical Internet Research, 22(5), e19458. McKendry, R. A., Rees, G., Cox, I. J., Johnson, A., Edelstein, M., Eland, A., ... & Heymann, D. (2020). Share mobile and social-media data to curb COVID-19. Nature, 580(7801), 29.

 >> Response: Thank you for your comment. Twitter has been used in research for health infodemiology studies since it has become popular, especially during the COVID-19 pandemic. Utilising instant data from social media is a contemporary method which could particularly benefit health professionals and public policy makers.

  1. It is also not clear what this methodological approach adds. The authors claim to follow SRQS which suggests that “The purpose of qualitative research is to understand the perspectives/experiences of individuals or groups and the contexts in which these perspectives or experiences are situated”. This paper does not do this.

 >> Response: we have checked to make sure that we have not missed any items in the SRQR guidelines. This guidelines is now being included as a new appendix table A. In addition, we have further demonstrated and explained our theory generated from the themes we identified from analysing the Tweets.

“This study follows the Standards for Reporting Qualitative Research (SRQR) guidelines (see Appendix A).”

“Discussion: We theorised the “Snowball effect” on social media is accounted for the mass-anxiety behaviour of panic buying toilet paper. The spontaneous contagion of fear and panic through social media could fuel psychological reactions in midst of crises. The popularity of negative and sentimental social media posts regarding the sufficiency of toilet paper acts as an emotional trigger of public anxiety and panic. “

“The popularity and influence of negative tweets could potentially lead to the “snowball effect” in social media, as regular social media users effortlessly pick up sentimental feelings and opinions from Tweets by others (Boulianne, 2015). Users are more likely to assimilate fearful emotions of running out of toilet paper from these Tweets, which could fuel anxiety-driven panic buying behaviours.”

1.15      Discussion: no theoretical framework is deployed in this paper. The researchers have a background in psychology and epidemiology, and could add some really interesting insights to this paper. I would like to see a much more in-depth analysis / discussion with insights / interpretations that are rooted in psychology / epidemiology.

 >> Response: We have revised our paper by adding content in the discussion with insights and interpretations, e.g. we discussed that the “snowball effect” is likely led by the popularity of panic buying-related tweets, as social media users could seamlessly assimilate fearful emotions of running out of toilet paper, which would influence their behaviours. We have provided additional explanations in this revised version which demonstrated clearer links of our findings to the theories we proposed, e.g.:

  • People believe they need more toilet paper during the pandemic because of the lockdown and staying at home protocols, work from home
  • The shortage of toilet paper is more visible than other goods in the supermarket, as toilet paper is bulky and occupied a lot of space,
  • Exposure to the spread of photos and Tweets of empty shelves of toilet paper on social media fuelled panic buying behaviours, which was particularly seen with a lot more hashtags about toilet paper shortage than other items, such as #toiletpapercrisis and #toiletpapergate

In addition, anxiety and panic are common symptoms during this pandemic period and we have added content on the psychological and neurological aspects to the article to highlight the importance of understanding panic buying behaviour. Please see the first paragraph of the Introduction, and the first and second paragraph of Discussion.

“From our analysis, close to half of the Tweets expressed negative feelings towards the panic buying of toilet paper, such as anger and frustration over the insufficient stock of toilet rolls from the stores. The top three most influential Tweets had a total of 588,000 likes and 80,000 Retweets, all of which were negative, expressing either confusion or anger towards the panic buying phenomena. We theorised that the popularity and influence of negative tweets could potentially lead to a “snowball effect” on social media, as regular social media users pick up sentimental feelings and opinions from Tweets by others (Boulianne, 2015). This is enhanced by the belief that they would consume more toilet paper than normal due to the lockdown and work-from-home restrictions. In addition, unlike canned food, pasta or masks and hand sanitisers, toilet paper is not an essential necessity for the pandemic. However, empty shelves of toilet paper are more noticeable in supermarkets, as toilet papers are bulkier and take up more space than perishable items. The exposure to these photos on social media acts as an emotional trigger of anxiety and panic of running out of toilet paper. Social media users are more likely to assimilate fearful emotions from these Tweets, which could fuel anxiety-driven panic buying behaviours.”

We thank the reviewer for their comments and suggestions.

Reviewer 2 Report

Dear Authors,

Thank you for the original paper, in several places it made me sincerely laugh :)

However, I have a few comments regarding the presentation.

The introduction should provide more information on the topic selected. Shortage of toilet paper can make people angry, frustrated and confused but toilet paper is not necessary to survive. For example, in the U.S. there was a shortage of drinking water. In our country, specific foods from stores shelves disappeared for a quite extended period like meat, frozen and canned food, groats, but we have never suffered from a shortage of TP :) Shortage of face masks, gloves and other medical supplies touched all countries. Thus, the reasons why people in Australia decided to hoard toilet paper remain unclear.

In the Methods section, you state that "We applied a grounded theory approach to develop themes and theories from the dataset using an interpretivist paradigm" but the theory is not presented. If I understand accurately, themes and categories should become the basis of a hypothesis or a new theory.

The text in the original Tweets contains swear-words. I have no previous experience, and I am not sure if it is acceptable for a scientific paper.

Author Response

Response to Reviewer 2’s Comments

Reviewer 2

Thank you for the original paper, in several places it made me sincerely laugh :)

>> Response:  Thank you for your kind words, we are glad that you found our paper delightful.

However, I have a few comments regarding the presentation.

2.1        The introduction should provide more information on the topic selected. Shortage of toilet paper can make people angry, frustrated and confused but toilet paper is not necessary to survive. For example, in the U.S. there was a shortage of drinking water. In our country, specific foods from stores shelves disappeared for a quite extended period like meat, frozen and canned food, groats, but we have never suffered from a shortage of TP :) Shortage of face masks, gloves and other medical supplies touched all countries. Thus, the reasons why people in Australia decided to hoard toilet paper remain unclear.

>> Response: We agreed that more context on the background information of toilet paper shortage could benefit readers' understanding on the situation. Anxiety-driven panic buying of toilet paper has been reported by mainstream media since March, 2020 in several countries, including locations in the U.S., Australia, Canada, the U.K., Singapore, Japan, Hong Kong and other cities. It was a widespread phenomenon on social media apart from the panic-buying of other perishable items and medical supplies. Therefore, it was interesting to further investigate the conversations around this social phenomenon.

“The hoarding of a particular household item, toilet paper, has occurred in several countries, including the U.S., Australia, Canada, the U.K., Singapore, Japan, Hong Kong and other cities (Jankowicz, 2020). This widespread phenomenon received a lot of public attention in news and social media, sending shockwaves to demand and supply chains around the globe (Kirk & Rifkin, 2020).”

2.2        In the Methods section, you state that "We applied a grounded theory approach to develop themes and theories from the dataset using an interpretivist paradigm" but the theory is not presented. If I understand accurately, themes and categories should become the basis of a hypothesis or a new theory.

>> Response: We agree that the theory developed from the qualitative coding was not made salient in the paper. The themes observed within the study were developed into the theory of a ‘snowball effect’. We have now made this salient in the paper:

“Discussion: We theorised the “Snowball effect” on social media is accounted for the mass-anxiety behaviour of panic buying toilet paper. The spontaneous contagion of fear and panic through social media could fuel psychological reactions in midst of crises. The popularity of negative and sentimental social media posts regarding the sufficiency of toilet paper acts as an emotional trigger of public anxiety and panic. “

“The popularity and influence of negative tweets could potentially lead to the “snowball effect” in social media, as regular social media users effortlessly pick up sentimental feelings and opinions from Tweets by others (Boulianne, 2015). Users are more likely to assimilate fearful emotions of running out of toilet paper from these Tweets, which could fuel anxiety-driven panic buying behaviours.”

2.3        The text in the original Tweets contains swear-words. I have no previous experience, and I am not sure if it is acceptable for a scientific paper.

>> Response: After revision, we have decided to replace letters in the swear words with asterisks.

We thank the reviewer for their comments and suggestions.

Reviewer 3 Report

The topic is current.

1. Anxiety and Panic are common symptoms during this pandemic period. Can the author introduce some more psychiatric or neurological references in the discussion? Please added these two important papers:      - Montemurro N. The emotional impact of COVID-19: From medical staff to common people. Brain Behav Immun. 2020 Jul;87:23-24. doi: 10.1016/j.bbi.2020.03.032.               - Di Carlo DT et al. Exploring the clinical association between neurological symptoms and COVID-19 pandemic outbreak: a systematic review of current literature. J Neurol. 2020 Aug 1:1–9. doi: 10.1007/s00415-020-09978-y. Epub ahead of print. PMID: 32740766; PMCID: PMC7395578.

2.

Figure 1 must be removed.

3.

Population is 25-35 aged. Do you find differences in people younger or older than 30 ?

4.

What about the role of e-commerce at this time also in this commercial sector?

Add these two important references in the discussion section:  - Gao X et al. To buy or not buy food online: The impact of the COVID-19 epidemic on the adoption of e-commerce in China. PLoS One. 2020 Aug 20;15(8):e0237900. doi: 10.1371/journal.pone.0237900.      - Will COVID-19 change neurosurgical clinical practice? Br J Neurosurg. 2020 Jun 1:1-2. doi: 10.1080/02688697.2020.1773399. Epub ahead of print. 

Author Response

Response to Reviewer 3’s Comments

Reviewer 3

The topic is current.

3.1        . Anxiety and Panic are common symptoms during this pandemic period. Can the author introduce some more psychiatric or neurological references in the discussion? Please added these two important papers:      - Montemurro N. The emotional impact of COVID-19: From medical staff to common people. Brain Behav Immun. 2020 Jul;87:23-24. doi: 10.1016/j.bbi.2020.03.032.    - Di Carlo DT et al. Exploring the clinical association between neurological symptoms and COVID-19 pandemic outbreak: a systematic review of current literature. J Neurol. 2020 Aug 1:1–9. doi: 10.1007/s00415-020-09978-y. Epub ahead of print. PMID: 32740766; PMCID: PMC7395578.

>> Response: Thank you for your suggestion. The authors agree that introducing psychological and neurological aspects to the article would highlight the importance of understanding panic buying behaviour. Please see the first paragraph of the Introduction, and the first paragraph of Discussion. The recommended references were also cited.

“An increasing body of evidence suggests that many patients infected with COVID-19 have developed neurological symptoms (Di Carlo, 2020).”

“The uncertainties about the duration of the pandemic and how it would affect our lives inflict profound psychological impacts on people (Montemurro, 2020)”

3.2.       Figure 1 must be removed.

>> Response: Thank you for your suggestion. This is now removed.

 3.3.  Population is 25-35 aged. Do you find differences in people younger or older than 30 ?

>> Response: Thank you for your comment. We believe that there was confusion over the statement regarding age. The statement (page 8, section “Researcher Characteristics”) describes the age of the researchers involved in this study because it was an item in the reporting guidelines we used. Age of the participants who posted the Tweets was not collected or analysed because our protocol of data analysis removed user information from the dataset before analysis. We have now revised our manuscript to make it clearly that the 25-35 is the age range of the researchers.

“Researcher characteristics

The researchers that carried out this research are based in Australia, aged 25-35, with background in psychology, epidemiology, and statistics. The researchers are not, and do not have acquaintances, who are prolific social mediasts.”

3.4.       What about the role of e-commerce at this time also in this commercial sector?

Add these two important references in the discussion section:  - Gao X et al. To buy or not buy food online: The impact of the COVID-19 epidemic on the adoption of e-commerce in China. PLoS One. 2020 Aug 20;15(8):e0237900. doi: 10.1371/journal.pone.0237900.      - Will COVID-19 change neurosurgical clinical practice? Br J Neurosurg. 2020 Jun 1:1-2. doi: 10.1080/02688697.2020.1773399. Epub ahead of print.

>> Response: Thank you for your suggestion. The authors agree that the suggested paper “Will COVID-19 change neurosurgical clinical practice” should be mentioned in our paper about the use of digital mental health support. Please see the third paragraph of the Discussion.

 “The COVID-19 pandemic may be a good opportunity to explore the feasibility of digital health options (Montemurro, 2020)”

We thank the reviewers for their comments and suggestions.

Round 2

Reviewer 3 Report

Authors solved all criticisms. Well done.